# Lactate induces synapse-specific potentiation on CA3 pyramidal cells of rat hippocampus

**Gabriel Herrera-López, Ernesto Griego[ID], Emilio J. Galván[ID]***

Departamento de Farmacobiología, Cinvestav Sede Sur, México City, México

* ejgalvan@cinvestav.mx

**Data Availability Statement:** The data underlying the results presented in the study are available from: https://gin.g-node.org/gabriel.helo/Lactate-potentiation

## Abstract

Neuronal activity within the physiologic range stimulates lactate production that, via metabolic pathways or operating through an array of G-protein-coupled receptors, regulates intrinsic excitability and synaptic transmission. The recent discovery that lactate exerts a tight control of ion channels, neurotransmitter release, and synaptic plasticity-related intracellular signaling cascades opens up the possibility that lactate regulates synaptic potentiation at central synapses. Here, we demonstrate that extracellular lactate (1–2 mM) induces glutamatergic potentiation on the recurrent collateral synapses of hippocampal CA3 pyramidal cells. This potentiation is independent of lactate transport and further metabolism, but requires activation of NMDA receptors, postsynaptic calcium accumulation, and activation of a G-protein-coupled receptor sensitive to cholera toxin. Furthermore, perfusion of 3,5-dihydroxybenzoic acid, a lactate receptor agonist, mimics this form of synaptic potentiation. The transduction mechanism underlying this novel form of synaptic plasticity requires G-protein βγ subunits, inositol-1,4,5-trisphosphate 3-kinase, PKC, and CaMKII. Activation of these signaling cascades is compartmentalized in a synapse-specific manner since lactate does not induce potentiation at the mossy fiber synapses of CA3 pyramidal cells. Consistent with this synapse-specific potentiation, lactate increases the output discharge of CA3 neurons when recurrent collaterals are repeatedly activated during lactate perfusion. This study provides new insights into the cellular mechanisms by which lactate, acting via a membrane receptor, contributes to the memory formation process.

## Introduction

Lactate, the final product of glycolysis, has gained recognition as a metabolic intermediate with multiple functions in brain physiology. Lactate is the most abundant monocarboxylate in the brain and has a central role in energy production. According to the astrocyte-to-neuron lactate shuttle hypothesis [1], astrocytes produce and release lactate into the extracellular interstice, which is then transported and oxidized by neurons to fuel synaptic transmission, a process that is highly energy demanding [2]. Furthermore, neural activity within the physiologic range stimulates lactate production [3, 4] and increases the lactate level up to 150% above the resting extracellular concentration [5].

A growing body of evidence shows that lactate is implicated in complex physiologic processes such as fear [6], ventilation [7], and the pathophysiology of depression [8]. Lactate also

**Funding:** GHL, fellowship 426505 Consejo Nacional de Ciencia y Tecnología, Mexico. EG, fellowship 727269 Consejo Nacional de Ciencia y Tecnología, Mexico. EJG, Grant CB-2016-281617 Consejo Nacional de Ciencia y Tecnología, Mexico.

**Competing interests:** The authors have declared that no competing interests exist.

modulates synaptic plasticity and memory formation. In this context, extracellular lactate supports hippocampal neuronal activity [9, 10] and allows induction of LTP in CA1 neurons [11]. Likewise, lactate plays a central role in long-term memory formation and maintenance of *in vivo* LTP [12, 13].

Since the discovery of hydroxycarboxylic acid receptors (HCA) in the brain, a family of seven transmembrane receptors primarily activated by lactate and other monocarboxylates [14], different groups have focused on understanding the role of lactate acting via these receptors. Thus far, the most studied is HCA1 receptor, which couples to inhibitory $G_{i/o}$ proteins and reduces neuronal excitability [15–17]. However, in the locus coeruleus, activation of lactate receptors is associated with intracellular calcium accumulation and increased excitability via activation of $G_s$ proteins [18]. In heterologous systems, the expression of the HCA1 receptor activates PKC, ERK ½, and the inositol-1,4,5-trisphosphate 3-kinase (IP3-K)/Akt pathway in a G-protein βγ-dependent manner [19]. Remarkably, activation of these intracellular cascades is involved in multiple forms of hippocampal plasticity.

The actions of lactate are complex and involve metabolic-dependent functions and transduction mechanisms via activation of membrane receptors. The latter process has been less investigated. Here we report the effects of lactate on the synaptic transmission of CA3 pyramidal cells (CA3 PC). Lactate triggers synaptic potentiation at the recurrent collateral synapses through a mechanism that primarily involves a postsynaptic lactate receptor, calcium accumulation, and NMDAR activity. At the intracellular level, potentiation requires activation of a G-protein βγ subunits, and the intracellular effects of IP3-K, PKC, and CaMKII. Because synaptic potentiation is restricted to the recurrent collateral synapses and absent at the mossy fiber synapses, we conclude that lactate triggers an input-specific form of synaptic plasticity on CA3 PC of the hippocampus.

## Material and methods

The Ethics Committee for Animal Research (CICUAL) of The Center for Research and Advanced Studies of the National Polytechnic Institute (CINVESTAV) approved the experimental procedures that mandate all efforts to minimize animal suffering (Protocol number 0090–14) according to Mexican legislation (NOM-062-ZOO-1999), which were performed in adherence to the "Guide for the care and use of Laboratory Animals (NIH 8th edition, 2011).

### Animals and acute slice preparation

We used male Sprague-Dawley rats (21–28 PND) housed in our local facilities with *ad libitum* access to food and water and under a normal light/dark cycle (12 h/12 h) in controlled temperature conditions (22˚C). Animals were anesthetized with sodium pentobarbital (70 mg/kg) and brains were rapidly removed and placed into ice-cold sucrose-based solution containing (mM): 210 sucrose, 2.8 KCl, 2 $MgSO_4$, 1.25 $Na_2HPO_4$, 25 $NaHCO_3$, 1 $MgCl_2$, 1 $CaCl_2$, and 10 D-glucose. The solution was continuously bubbled with carbogen (95% $O_2$/ 5% $CO_2$). Tissue blocks containing the hippocampus and surrounding structures were then sliced (385 μM) in the transversal plane using a vibratome (Leica VT1000S). The resulting slices were placed into a holding solution for 30 min at 32–34˚C, and then kept for 90 min at room temperature before any experimental procedure. The holding solution consisted of a modified aCSF containing (mM): 125 NaCl, 2.5 KCl, 1.25 $Na_2HPO_4$, 25 $NaHCO_3$, 4 $MgCl_2$, 1 $CaCl_2$, and 10 D-glucose, bubbled with carbogen.

At the end of the incubation period, individual slices were transferred to a recording chamber (total volume 400 μl, maintained at 33˚C) and continuously perfused with aCSF containing

(mM): 125 NaCl, 2.5 KCl, 1.25 $Na_2HPO_4$, 25 $NaHCO_3$, 2 $MgCl_2$, 2 $CaCl_2$, and 10 D-glucose, bubbled with carbogen to a final pH of between 7.35 and 7.4.

## Electrophysiological recordings

We performed voltage- and current-clamp whole-cell recordings from CA3 PC with an Axopatch 200B amplifier (Molecular Devices, Palo Alto, CA, USA), digitized and sampled at 10 kHz and filtered at 5 kHz (Digidata 1440A; Molecular Devices). Signals were acquired and off-line analyzed with pCLAMP10 software (Molecular Devices). The cells were visually identified with differential interference contrast optics coupled to a Nikon FN1 microscope. We used borosilicate pipettes with a final resistance of 3–5 MΩ filled with a solution containing (mM): 120 $Cs^+$methanesulfonate, 10 KCl, 0.5 EGTA, 10 HEPES, 4 $Mg^{2+}$-ATP, 0.3 $Na^+$-GTP, 8 phosphocreatine; pH = 7.2–7.26, adjusted with HCl when necessary. In some experiments, BAPTA tetrapotassium salt (20 mM) was included in the pipette, thus the $Cs^+$methanesulfonate concentration was adjusted to 75 mM to reach a final osmotic pressure ≈300 mOsm. For current-clamp experiments, the intracellular solution composition was (mM): 135 $K^+$- gluconate, 10 KCl, 10 HEPES, 1 $MgCl_2$, 2 ATP disodium hydrate, 0.3 $Na^+$-GTP, 8 phosphocreatine; pH = 7.2–7.26, adjusted with HCl.

## Stimulation techniques

We applied monopolar current pulses via a bipolar electrode localized in the stratum radiatum of CA3 and the superior blade of the dentate gyrus to evoke excitatory postsynaptic currents (EPSC) from recurrent collateral (RC) and mossy fiber synapses (MF), respectively. For each pathway, we injected a maximum of 400 ± 55 μA of current using a stimulus isolator (A360, WPI). Pairs of stimuli were delivered at 70 ms interval (interstimulus interval [ISI]) with a frequency of 0.06 Hz. After stabilization of the synaptic response, we recorded the baseline activity in voltage-clamp mode for at least 10 minutes before continuing with the perfusion of different drugs. For tandem experiments of converging MF and RC responses on a CA3 PC (Fig 1), MF- and RC EPSCs were consecutively acquired on the same sweep with a difference of 1000 ms, and the ISI between sweeps was set to 15 sec (0.06 Hz) to minimize synaptic interactions.

The experiments were systematically performed in the presence of picrotoxin (PTX, 50 μM), an antagonist of $GABA_A$ receptors, to isolate the glutamatergic response. At the end of the recording period, (2S,2'R,3'R)-2-(2',3'-dicarboxycyclopropyl)glycine (DCG-IV, 5 μM), an agonist of the group II/III metabotropic glutamate receptors mGluRs expressed in the mossy fiber terminals, but not in CA3 pyramidal neurons [20, 21], was included in the perfusate to confirm the origin of the synaptic response.

For current-clamp experiments, we recorded 20 consecutive voltage sweeps consisting of 10 stimuli applied at 20 and 40 Hz using the minimal stimulation current to evoke a spike probability near to zero. The cells were maintained at their respective resting membrane potential (≈ -70 mV) without any block of the GABAergic transmission.

## Drugs

All chemicals and drugs were obtained from Sigma-Aldrich Chemicals Co. (St. Louis, MO, USA) and freshly prepared directly in the required solution. Sodium L-lactate was added to the aCSF, and the NaCl concentration was adjusted to prevent changes in the osmolarity. The inclusion of L-lactate did not induce changes in the pH of the aCSF (≈7.35–7.4). Gallein, α-cyano-4- hydroxycinnamic acid (4-CIN), U73122, wortmannin, and chelerythrine were directly diluted in DMSO to obtain a stock solution and then diluted in aCSF or intracellular

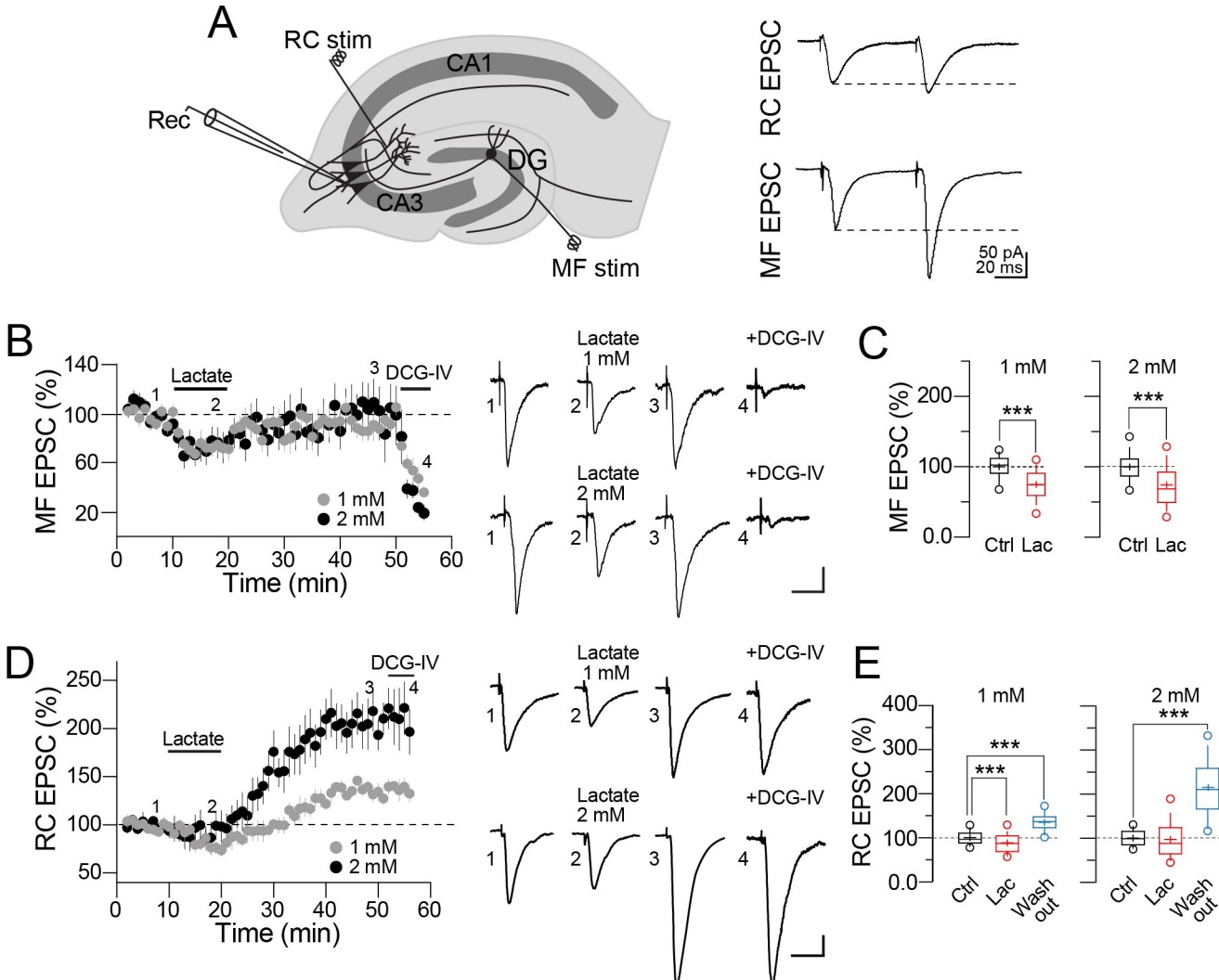

**Fig 1. Lactate triggers synaptic potentiation at the recurrent collateral synapse of CA3 pyramidal cells. A)** Schematic representation of the hippocampal slice showing the electrode array area. Mossy fibers were stimulated (MF stim) in the superior blade of the dentate gyrus (DG), whereas recurrent collaterals (RC stim) were stimulated in the stratum radiatum of area CA3. Right panel, representative CA3 PC responses. RC EPSC exhibited shorter stimulation latencies (≈1.5–2 ms) and larger EPSC decay time constants compared to MF EPSC that exhibited strong paired-pulse facilitation, larger stimulation latencies (≈2–3 ms) and were depressed in response to activation of group II mGluRs with DCG-IV. The tandem acquisition of the evoked responses was performed with 1 s delay between MF and RC stimulation and the whole-cell recordings were restricted to PC located in the CA3b section of the hippocampal slice. **B)** Average time-course graph showing the effects of lactate (1 or 2 mM; gray and solid symbols, respectively) on the MF EPSC. The solid bars indicate the perfusion of lactate and DCG-IV, respectively. The right panel shows representative MF EPSC traces (averaged from 5 continuous sweeps) in control (1), during perfusion of lactate (2), 30 min after washout of lactate (3), and in the presence of DCG-IV. **C)** Box plots summarizing the actions of lactate on the MF EPSC (Mann-Whitney U test, *p < 0.05 vs baseline). **D)** Average time-course graph showing the effects of lactate (1 or 2 mM; gray and solid symbols, respectively) on the RC EPSC. The representative RC EPSC traces (averaged from five continuous sweeps) show the lactate effects at the two concentrations tested. **E)** Box plots summarizing the actions of lactate on the RC EPSC (One-way ANOVA on ranks, Tukey, *p < 0.05; ***p < 0.001). Calibration bars for panel A, 50 pA / 20 ms; for panel B 25 pA / 25 ms; for panel D 50 pA / 25 ms.

solution (as needed) to the required concentration. DCG-IV was purchased from Tocris (Bristol, UK). The final concentration of DMSO was always below 0.5%. The aliquots prepared in DMSO were stored at -80˚C and used within 1 week of preparation. Pertussis toxin (Millipore, Burlington, MA) was diluted in glycerol as a stock solution.

## Statistical analysis

Group measures are expressed as the normalized mean of the current amplitude ± SEM (unless otherwise indicated). Assessment of statistical differences between means was performed with Mann-Whitney U test, and for multiple comparisons one repeated measures ANOVA on ranks, followed by a post-hoc test of Tukey. In all tests, the significance level was set at $p < 0.05$.

# Results

## Lactate triggers synapse-specific potentiation of the excitatory postsynaptic currents of CA3 pyramidal cells

We examined the effect of lactate on the EPSC of CA3 PC evoked on the mossy fibers (MF EPSC) and recurrent collateral synapses (RC EPSC) (Fig 1A). First, we performed a tandem analysis to determine the effects of lactate on the synaptic transmission converging on CA3 PC. MF and RC responses were successively acquired (see methods for details). To isolate the glutamatergic transmission, the experiments were performed in the presence of picrotoxin (50 μM), unless otherwise specified. After collecting a steady 10 min baseline of MF and RC EPSC, we perfused lactate (1 mM or 2 mM) for 10 min, followed by a washout that continued for up to 35 min. As summarized in the averaged time-course graph (Fig 1B), perfusion of lactate transiently depressed the MF EPSC; upon washout, the MF EPSC returned to its baseline value (MF EPSC in the presence of lactate 1 mM = 74.2 ± 2.7% of baseline response, n = 8; in the presence of lactate 2 mM = 73.3 ± 3.4% of baseline response, n = 10; Fig 1B and 1C). To verify the MF origin of the synaptic response, we perfused the group II/III mGluRs agonist, DCG-IV (5 μM) [20, 21]. As expected, perfusion of DCG-IV depressed the evoked response (MF EPSC in the presence of DCG-IV = 9.1 ± 3.4% of baseline response).

Contrary to this observation, the RC synapse showed a different response to lactate perfusion. As illustrated (Fig 1D), lactate triggered a biphasic effect. An initial moderate depression of the RC EPSC was followed by a sustained enhancement of the RC EPSC amplitude, insensitive to perfusion of DCG-IV (RC EPSC at 30 min after washout of 1 mM lactate = 135.6 ± 2.4% of baseline response, $p < 0.05$, n = 7; gray symbols. Fig 1D). In experiments in which 2 mM lactate was perfused, we observed a marked, long-lasting enhancement of the RC EPSC amplitude (RC EPSC at 30 min after lactate washout = 215.12 ± 7.5% of baseline response, $p < 0.05$, n = 9; solid symbols, Fig 1D). The effect of both lactate concentrations is summarized in the time-course graph (Fig 1D) and boxplots (Fig 1E). Because previous studies established that neuronal activity leads to a rise in extracellular lactate that fluctuates at around 2 mM [4], we used this lactate concentration. Also, the previous experiments were performed with an aCSF containing a higher glucose concentration (10 mM). Thus, in another group of cells, we investigated if under a less glucose-saturated condition, lactate is capable of potentiating the synaptic transmission. Thus, glucose was decreased to 5 mM or was removed from the aCSF. In slices perfused with glucose 5 mM, we found that lactate (2 mM) potentiates the RC EPSC to a similar extent (RC EPSC at 30 min after lactate washout = 195.22 ± 14.5% of baseline response, $p < 0.01$, n = 3; S1A Fig). Contrary to this observation, we were unable to maintain healthy brain slices nor stable synaptic responses when glucose was lowered to 2 mM or was removed from the aCSF (S1B–S1D Fig). Collectively, this first set of experiments demonstrates for the first time that an extracellular increase in lactate triggers a long-lasting form of synaptic potentiation on the RC synapse of CA3 PC, whereas at MF synapses synaptic transmission is transiently depressed and no synaptic potentiation is initiated.

## Lactate modifies the paired-pulse ratio and coefficient of variation of CA3 pyramidal cells

Because the previous responses were evoked using paired pulses (ISI = 70 ms), we also monitored the paired-pulse ratio (PPR) and coefficient of variation ($CV^{-2}$) on baseline conditions, during lactate perfusion and washout. These analyses aimed to identify if the extracellular lactate involves a presynaptic locus since changes in the $CV^{-2}$ indicate modulation of the presynaptic terminal [22]. We found that during lactate perfusion the MF PPR transiently decreased and returned to baseline value during lactate washout (MF PPR at baseline = 2.2 ± 0.2; during lactate perfusion = 1.5 ± 0.14, p<0.01; washout = 2.1 ± 0.3; n = 10; Fig 2A). Consistent with this observation, lactate reduced the MF $CV^{-2}$ ($CV^{-2}$ during perfusion of lactate = 46.06 ± 8.9%; $CV^{-2}$ at washout = 119.8 ± 17%; Fig 2B). On the RC synapse, lactate transiently increased the PPR, which returned to baseline during washout (RC PPR at baseline = 1.3 ± 0.12; during lactate perfusion = 1.88 ± 0.2, p<0.05; washout = 1.32 ± 0.23; n = 9; Fig 2C). Likewise, perfusion of lactate caused a transient decrease in the RC $CV^{-2}$, which returned to baseline value during washout ($CV^{-2}$ during perfusion of lactate = 45.9 ± 8.5%; $CV^{-2}$ at washout = 97.5 ± 8.7%; Fig 2D). Taken together, the changes in the PPR and $CV^{-2}$ suggest that transient surges in extracellular lactate, exert presynaptic control of neurotransmitter release. Because potentiation of the RC synapse is not accompanied by a persistent change in PPR or $CV^{-2}$, our experiments also suggest the additional requirement of a postsynaptic mechanism to trigger synaptic potentiation.

## Postsynaptic requirements for synaptic potentiation

To explore if this novel form of synaptic potentiation also requires activation of postsynaptic components, we voltage clamped CA3 PC at -100 mV during lactate perfusion. If synaptic potentiation requires a voltage-dependent postsynaptic mechanism, the voltage inactivation should prevent potentiation mediated by lactate. We tested this prediction by continuously maintaining the recorded cell at -100 mV during lactate perfusion and releasing the voltage-clamping to the CA3 PC RMP (≈ -70 mV) during lactate washout. This voltage manipulation

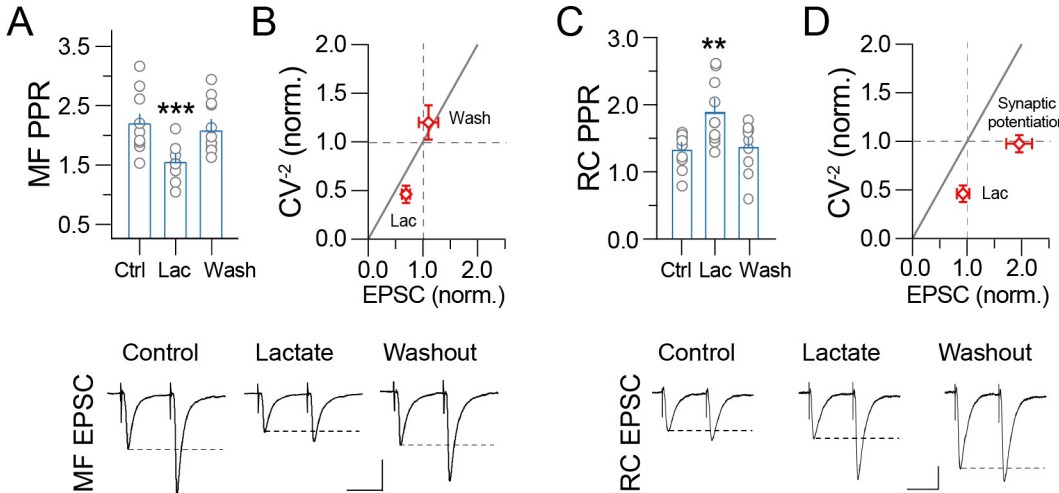

**Fig 2. Presynaptic modifications triggered by lactate perfusion.** Analysis of the paired-pulse ratio and $CV^{-2}$ of the **A–B)** MF EPSC and **C–D)** RC EPSC in control conditions, during lactate (2 mM) perfusion, and at 30 min lactate washout. Lactate perfusion shifts the $CV^{-2}$ below the identity lines (horizontal and diagonal lines), indicating presynaptic action. The lower panels show representative paired-pulse EPSC evoked with an ISI of 70 ms (One-way ANOVA on ranks, Tukey, *p < 0.05). Calibration bars, 50 pA / 50 ms.

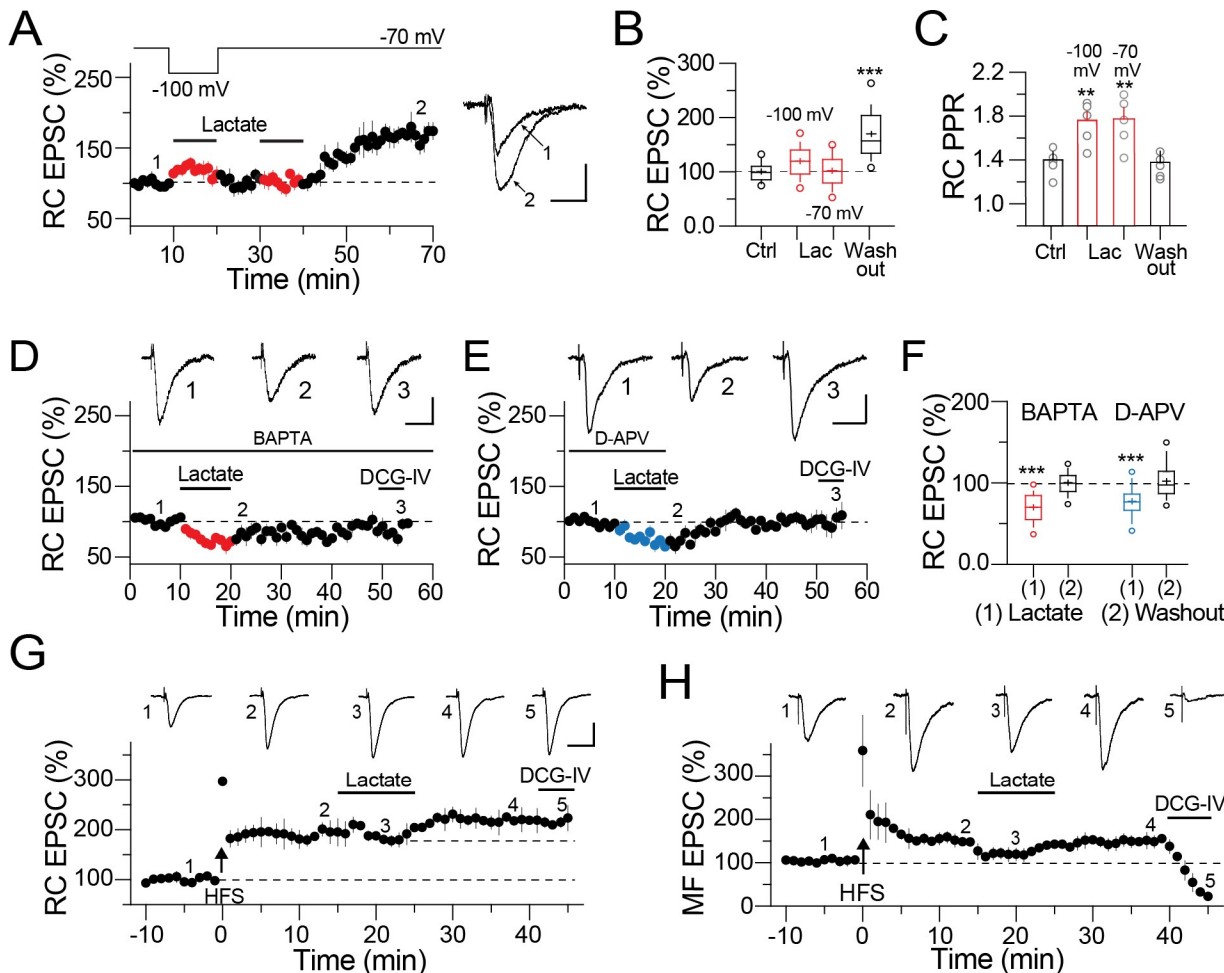

**Fig 3. Postsynaptic requirements for synaptic potentiation of CA3 pyramidal cells. A)** Averaged time-course graph; lactate (2 mM) was perfused while cells were held at -100 mV (red symbols). 5 min after washout, lactate perfusion with cells at -70 mV induced synaptic potentiation. Inset, representative RC EPSC obtained in control conditions (1) and 30 min after lactate washout (2). **B)** Boxplots summarizing the changes in the RC EPSC amplitude during baseline (solid box), lactate perfusion at -100 mV and -70 mV (red boxes) and at 30 min washout (solid line; One-way ANOVA on ranks, Tukey, ***p<0.001). **C)** RC PPR analysis at the indicated conditions (One-way ANOVA on ranks, Tukey, *p<0.05). **D)** Averaged time-course graph showing the effects of lactate in cells loaded with BAPTA (20 mM). The upper traces are representative RC EPSC (averaged from five continuous sweeps, acquired at the indicated time in the time-course graph). **E)** Averaged time-course graph and representative traces showing the effect of lactate in the presence of D-APV (50 μM). **F)** Boxplots summarizing the changes in the RC EPSC amplitude during lactate perfusion and at 30 min for the experiments performed with BAPTA or D-APV (One-way ANOVA on ranks, Tukey, ***p < 0.001). **G)** Averaged time-course graph and representative RC EPSC (averaged from five continuous sweeps, acquired at the indicated time in the time-course graph) in response to SR HFS (100 Hz, repeated 3 times, 10 sec interval) showing that lactate (2 mM for 10 min) triggers additional synaptic potentiation. **H)** Averaged time-course graph and representative MF EPSC (acquired at the indicated time in the time-course graph) in response to stratum lucidum HFS (100 Hz, repeated 3 times, 10 sec interval) showing that lactate (2 mM for 10 min) causes transient depression without affecting the magnitude of MF LTP upon lactate removal. Calibration bars for panel A, 20 pA / 25 ms; for panel D and E: 25 pA / 25 ms; for panel H and G, 50 pA / 50 ms.

prevented the amplitude enhancement of the RC EPSC during lactate perfusion. As illustrated (Fig 3A), at 10 min of lactate washout the amplitude of the evoked RC EPSC did not differ from that of control (RC EPSC amplitude = 101.5 ± 3.4%, p > 0.05). A second perfusion of lactate was applied while the recorded cell was maintained at -70 mV. Under this condition, the RC EPSC amplitude exhibited a sustained enhancement that continued up to 30 min after lactate washout (RC EPSC amplitude at 30 min washout = 169.8 ± 5.6%, p < 0.05 vs baseline; n = 9; Fig 3A and 3B). Consistent with these results, perfusion of lactate, independent of the

holding level of the postsynaptic cell, increased the PPR. The PPR returned to baseline during washout (RC PPR control = 1.41 ± 0.07; during lactate at -100 mV = 1.97 ± 0.07; during lactate at -70 mV = 1.84 ± 0.1; at 30 min washout = 1.44 ± 0.05; Fig 3C). Together, the lack of potentiation during the silencing of the voltage-dependent postsynaptic components and the concomitant potentiation when the postsynaptic cell is available for voltage-dependent activity, indicate that induction of synaptic potentiation by lactate requires postsynaptic activity.

Next, we investigated a series of potential components located at the postsynaptic compartment that may participate in the synaptic potentiation mediated by lactate. First, we explored the participation of postsynaptic calcium. Therefore, BAPTA (20 mM) was included in the patch pipette. After the seal break-in from gigaseal to whole-cell configuration, BAPTA was diffused for at least 15 min. Next, we acquired a baseline of RC EPSC followed by perfusion of lactate (2 mM), and washout for up to 30 min. The BAPTA loading prevented synaptic potentiation. Furthermore, we also observed a transient depression of the RC EPSC during lactate perfusion that returned to baseline value during washout (RC EPSC during lactate = 70 ± 2.8% of baseline response; RC EPSC at 35 min washout = 101.1 ± 1.02% of baseline response; n = 5; Fig 3D and 3F). Next, since the RC synapses of CA3 PC have a predominant NMDAR component involved in RC long-term potentiation [23], we investigated the role of NMDAR on the lactate-mediated synaptic potentiation. Thus, D-APV (50 μM) was bath applied during baseline and perfusion of lactate. We found that blockade of NMDAR also prevented induction of synaptic potentiation. During this pharmacologic condition, lactate depressed the RC EPSC, which returned to baseline value during washout (RC EPSC during lactate perfusion = 77 ± 2.6%; during washout = 101.14 ± 1.25; Fig 3E). Lastly, to determine whether the lactate-induced potentiation shares a common mechanism with the electrically-evoked LTP, occlusion experiments were carried out. Indeed, RC EPSC were isolated in the presence of picrotoxin (50 μM) and high frequency stimulation (HFS, 100 pulses at 100 Hz, repeated 3 times) was delivered to the CA3 SR. HFS induced robust EPSC enhancement (RC EPSC at 15 min post-HFS = 190.23 ± 3.67 of baseline response; p<0.001; Fig 3G). Then, lactate (2 mM for 10 min, solid bar in Fig 3G) was perfused at 15 min post-HFS. As illustrated, perfusion of lactate caused additional potentiation of the RC EPSC, that was insensitive to perfusion of DCG-IV (RC EPSC at 20 min after lactate washout = 218.86 ± 1.69 of baseline response, p<0.001, n = 3). In parallel experiments, the MF EPSC was isolated in the presence of D-APV (50 μM) + picrotoxin (50 μM) and HFS (100 pulses at 100 Hz, repeated 3 times) was delivered to the MF bundle. Notably, induction of MF LTP exhibited partial depression during perfusion of lactate (MF EPSC at 15 min post-HFS = 154.03 ± 1.72% of baseline response; p<0.001. During lactate perfusion = 121.97 ± 4.80% of baseline response, p<0.01). The partial depression was reverted during lactate washout and MF LTP returned to the previous potentiation level (MF EPSC at 25 min after washout of 2 mM lactate = 151.28 ± 1.39% of baseline response, p<0.001, n = 3; Fig 3H); the potentiated MF EPSC was sensitive to perfusion of DCG-IV (7.80 ± 6.22 of baseline response, p<0.001) corroborating its synaptic origin.

Taken together, these experiments indicate that lactate-mediated potentiation at the RC synapses of CA3 PC entails a voltage-dependent mechanism, activation of NMDAR and requires a transient calcium rise. Also, the enhanced RC response by lactate after electrically-evoked LTP, indicates that a transient rise in extracellular lactate is sufficient to enhance the synaptic strength of the RC synapse.

## Synaptic potentiation is independent of lactate metabolism

Extracellular lactate is taken up by neurons and transported to the cell body via the monocarboxylate transporter 2 (MCT2). Lactate is then catalyzed to pyruvate via the lactate

dehydrogenase (LDH) enzyme, and the resulting product has metabolic and neuronal functions. Thus, we explored whether metabolism is involved in the synaptic potentiation mediated by lactate.

First, we blocked the isomeric conversion of lactate to pyruvate by preincubating the slices with the non-competitive inhibitor of LDH, oxamate (10 mM, 10–20 min). Preincubation with oxamate did not interfere with synaptic potentiation when lactate (2 mM) was perfused (RC EPSC at 35 min of lactate washout = 207.2 ± 13.8%, p<0.05 vs baseline, n = 4; Fig 4A). However, oxamate preincubation did prevent the increase in the PPR during lactate perfusion (RC PPR control = 1.43 ± 0.05; during lactate perfusion = 1.41 ± 0.06, p = 0.57; S2A Fig). Consistent with this finding, in another group of CA3 PC we perfused pyruvate (2 mM) instead of lactate. Pyruvate (2 mM) increased, in a transient manner, the RC PPR (RC PPR control = 1.46 ± 0.03; during pyruvate perfusion = 1.97 ± 0.06; at 30 min washout = 1.52 ± 0.03; n = 3; p = <0.001, One-way ANOVA on ranks, Tukey, Fig 4B). These results support the notion that lactate metabolism exerts presynaptic modulation of glutamate release. To dissect the pre- and postsynaptic modulation that lactate exerts on synaptic transmission, we performed an arithmetic subtraction between the time-course response obtained when lactate was perfused alone (Fig 4C, solid line) and the response obtained from slices pretreated with oxamate and lactate (red line). The residual values of this subtraction (green line) show a transient

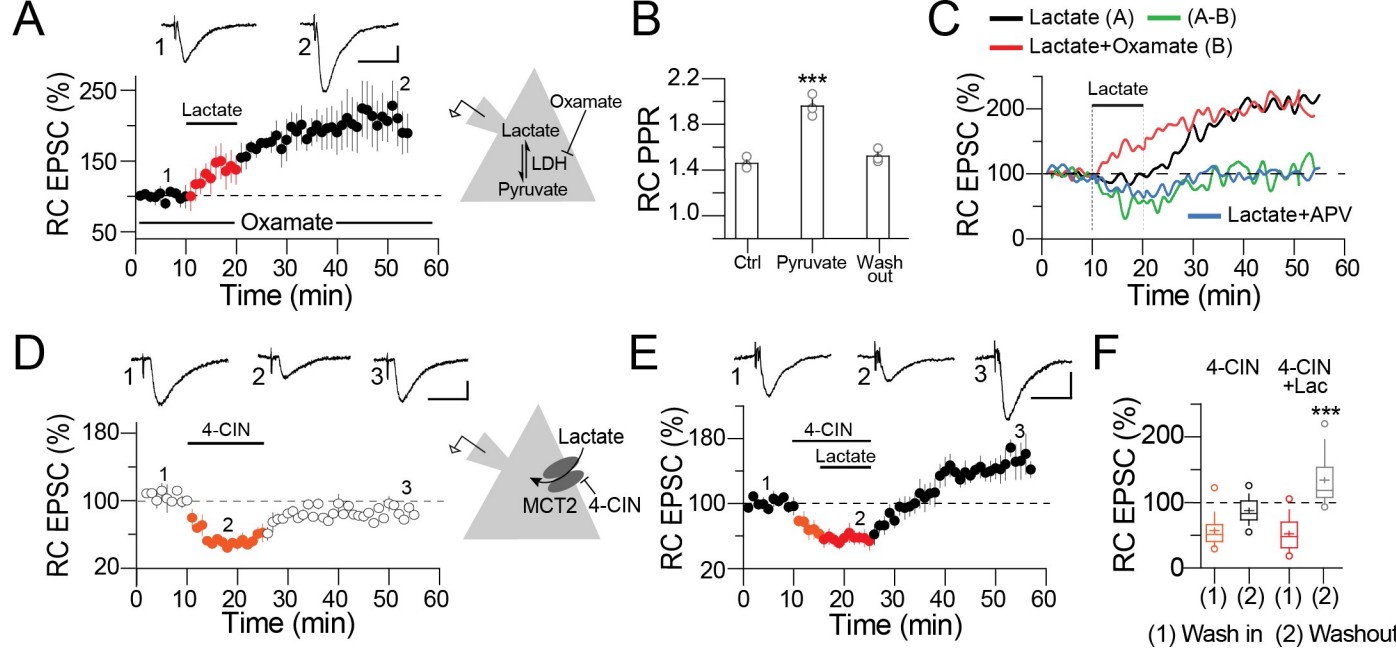

**Fig 4. Synaptic potentiation is independent of lactate metabolism. A)** Averaged time-course graph and representative current traces acquired at the indicated time of the time-course graph. Before the beginning of the experiment, the slices were preincubated (15–20 min) with the non-competitive blocker of the lactate dehydrogenase enzyme, oxamate (10 mM), to block the conversion of lactate to pyruvate. Oxamate did not interfere with synaptic potentiation induced with lactate. **B)** Bar graphs, Effect of pyruvate (2 mM) on the PPR of the RC EPSC. Similar to lactate, pyruvate exerts a transitory, presynaptic modulation of the synaptic transmission. (One-way ANOVA on ranks, Tukey, ***p < 0.001). **C)** Temporal sequence of the predictable presynaptic and postsynaptic contribution of lactate to synaptic potentiation. The solid line shows the time course of the normalized amplitude of the RC EPSC during baseline, lactate perfusion, and lactate washout. Red line, time course of the normalized RC EPSC response preincubated with oxamate. The subtraction of the normalized values of lactate minus the values obtained with oxamate (green line). This time course shows similar kinetics to those observed when lactate was perfused in the presence of D-APV (blue line). **D)** Averaged time-course graph and representative current traces acquired at the indicated time of the time-course graph showing the effect of 4-CIN (0.5 mM) on the RC EPSC. **E)** Averaged time-course graph and representative traces acquired at the indicated time showing the effects of 4-CIN and the subsequent perfusion of lactate (2 mM) on the RC EPSC. 4-CIN depressed the evoked response and further perfusion of lactate triggers synaptic potentiation. **F)** Boxplots summarizing the actions of 4-CIN or 4-CIN + lactate during perfusion of the drugs (wash in) and at 30 min (washout) on the RC EPSC amplitude (One-way ANOVA on Ranks, Tukey, *p < 0.001). Calibration bars: Panel A: 20 pA / 25 ms; panel B: 50 pA / 25 ms; panel C and D: 20 pA / 20 ms.

depression that returns to baseline value. This subtraction indicates both the metabolic contribution and the presynaptic modulation of glutamate release exerted by lactate. Interestingly, the kinetics of the residual plot exhibited a time-course similar to that observed in CA3 PC perfused with D-APV (blue line; see also Fig 3E), a pharmacologic treatment that blocks synaptic potentiation but unmasks the presynaptic contribution of lactate to modulate neurotransmitter release. Indeed, this manipulation corroborates the theory that lactate controls neurotransmitter release and triggers postsynaptic potentiation. We next blocked lactate uptake mediated by the MCT2, to rule out possible metabolic actions during synaptic potentiation. If lactate uptake and further metabolism are necessary for synaptic potentiation, blockade of the MCT2 should prevent potentiation. To test this hypothesis, we perfused the MCT2 blocker 4-CIN (0.5 mM). We first noticed that perfusion of 4-CIN alone transiently depressed the RC EPSC amplitude; upon 4-CIN washout, the synaptic response returned to baseline value (RC EPSC amplitude during 4-CIN perfusion = 53.8 ± 2.9% of baseline response, $p < 0.01$; at 30 min washout = 98.5 ± 4.5% of baseline response, n = 3; Fig 4D). Furthermore, the transient depression induced by 4-CIN did not modify the RC PPR (RC PPR during 4-CIN perfusion = 1.4 ± 0.06, n = 5; S2B Fig). These results show that lactate uptake via the MCT2 works as a non-glucose fuel for neuronal metabolism. 4-CIN was then present when lactate (2 mM) was applied. As summarized (Fig 4E), perfusion of 4-CIN decreased the amplitude of the RC EPSC. However, the subsequent perfusion of lactate triggered a sustained increase of the RC EPSC amplitude (RC EPSC during 4-CIN + lactate = 61.2 ± 3.6% of baseline response, $p < 0.05$; at 35 min washout = 137.9 ± 5.2%, $p < 0.05$; n = 5; Fig 4E and 4F). The combination of 4-CIN + lactate also prevented the increase in the PPR observed when lactate is perfused alone (RC PPR in the presence of 4-CIN + lactate = 1.5 ± 0.1, $p = 0.061$, n = 5; (S2B Fig). These experiments support the notion that synaptic potentiation mediated by extracellular lactate is initiated through a metabolic-independent pathway.

## Activation of a lactate receptor is required for synaptic potentiation

PC of the hippocampus express multiple HCA receptors, including the HCA1 or lactate receptor [16, 24]. Therefore, we explored the possible participation of HCA receptors in the synaptic potentiation mediated by lactate. We acquired a baseline of RC EPSC, followed by perfusion of the HCA1 agonist, 3,5-dihydroxibenzoic acid (3,5-DHBA, 0.5 mM, 10 min). Stimulation with 3,5-DHBA induced a persistent enhancement of the RC EPSC amplitude (RC EPSC at 30 min washout = 285.3 ± 11.5%, $p < 0.001$, n = 5; Fig 5A and 5B). Notably, 3,5-DHBA did not alter the PPR (RC PPR control = 1.36 ± 0.03; with 3,5-DHBA = 1.39 ± 0.03, $p > 0.05$; during washout = 1.37 ± 0.05; n = 5; Fig 5C), indicating postsynaptic localization of the receptor. Because the HCA1 is a $G_{\alpha i/o}$-coupled receptor [16, 25], we blocked the intracellular signaling events mediated by activation of the $G_{\alpha i/o}$ protein. For this, we included pertussis toxin (PT; 100 ng/ml) in the patch pipette. After the seal break-in from gigaseal to whole-cell configuration, PT was left to dialyze for at least 15–20 min. Unexpectedly, PT did not abolish the synaptic potentiation mediated by lactate perfusion (RC EPSC at 35 min washout = 199.88 ± 6.96%, $p < 0.001$, n = 8; Fig 5D and 5F). Also, the intracellular dialysis of PT did not alter the increase in the RC PPR during lactate perfusion (RC PPR control = 1.44 ± 0.09; during lactate perfusion = 2.05 ± 0.12, $p = 0.06$; during washout = 1.42 ± 0.07; n = 8; Fig 5G). Because our previous result indicates that lactate acts via a different signaling cascade, we explored the possible participation of stimulatory $G_s$-proteins, since a previous report found that lactate also activates a $G_s$-protein-coupled receptor [18]. Therefore, we blocked the $G_s$-dependent transduction mechanism by loading the patch pipettes with cholera toxin (CTX, 2.5 µg/ml). Similar to the previous experiment, CTX was dialyzed for 15–20 min before perfusion of lactate (2 mM).

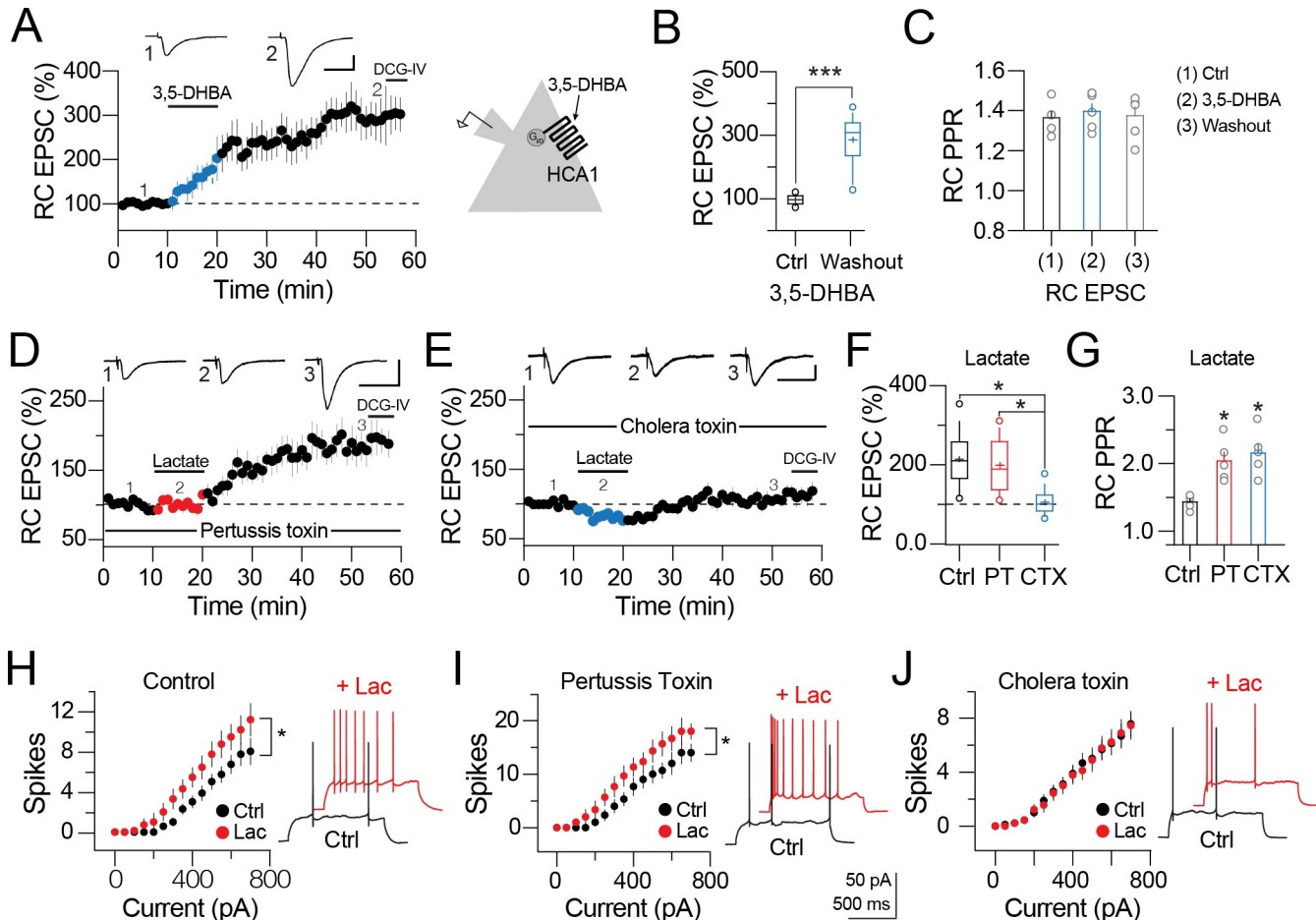

**Fig 5. Synaptic potentiation involves activation of Gₛ proteins. A)** Averaged time course and representative traces acquired at the indicated time showing the effects of 3,5-DHBA (0.5 mM) on the RC EPSC. Blue symbols represent 3,5-DHBA perfusion (10 min). **B)** Boxplots summarizing the magnitude of synaptic potentiation induced with 3,5-DHBA at 35 min of lactate washout. **C)** RC PPR analysis during baseline (1), perfusion of 3,5-DHBA (2), and at 35 min washout (3). **D)** Averaged time-course graph and representative traces acquired at the indicated time showing the effects of intracellular dialysis of pertussis toxin (PT, 100 ng/ml) and lactate perfusion (red symbols). **E)** Averaged time-course graph and representative traces from cells intracellularly loaded with cholera toxin (CTX, 2.5 μg/ml). CTX abolished the synaptic potentiation when lactate 2 mM was perfused (blue symbols). The synaptic responses from cells loaded with PT or CTX were insensitive to the application of DCG-IV 5 μM (solid bar on the time-course graphs). **F)** Boxplots summarizing the change on the RC EPSC amplitude during lactate perfusion in control, and in cells loaded with pertussis toxin or cholera toxin (red and blue boxes, respectively). **G)** RC PPR analysis in control conditions (solid bar), during lactate perfusion in cells loaded with pertussis toxin (red bar) or cholera toxin (blue bar) (One-way ANOVA on ranks, Tukey, *p<0.05). **H–J)** Scatter plots showing number of action potentials (spikes) elicited in response to current injections and the effects of lactate (2 mM) in control cells, cells loaded with pertussis toxin or cholera toxin. Cholera toxin prevented the increased number of spikes in response to lactate perfusion (Two-ways ANOVA on Ranks, Tukey, *p < 0.05).

Notably, under this experimental condition, lactate failed to induce synaptic potentiation (RC EPSC at 35 min = 107.6 ± 3.5%, n.s., n = 9; Fig 5E and 5F). Despite the lack of potentiation, we did observe a transient increase in the RC PPR that returned to baseline at 30 min of lactate washout (RC PPR control = 1.4 ± 0.1, during lactate perfusion = 2.16 ± 0.13, p<0.05; during washout = 1.39 ± 0.1. Fig 5G).

In a previous publication, we demonstrated that perfusion of lactate reduces the action potential discharge of CA1 PC [17]. Thus, to establish if lactate controls the output of CA3 PC via activation of a postsynaptic receptor, the firing discharge of CA3 PC was assessed before and after lactate perfusion. Unexpectedly, lactate increased the firing discharge of CA3 PC. As summarized (Fig 5H), under control conditions, the CA3 PC responded with a maximal AP

number of 8 ± 1.3 spikes. Perfusion of lactate (2 mM) increased the AP discharge to 11.1 ± 1.6 spikes (One-way ANOVA on Ranks, Tukey, p<0.05, n = 8; Fig 5H). The increased AP discharge in the presence of lactate was also observed in CA3 PC loaded with PT (baseline AP discharge = 14 ± 1.6 spikes; AP discharge in the presence of lactate [2 mM] = 18 ± 1.4 spikes, p<0.05; n = 5; Fig 5I). In contrast, the increased AP discharge induced with lactate was abolished in CA3 PC loaded with CTX (baseline AP discharge = 7.1 ± 1 spikes; AP discharge in the presence of lactate [2 mM] = 7 ± 1 spikes, n.s., n = 10; Fig 5J). These results demonstrate that CA3 PC express a lactate-sensitive receptor coupled to a $G_s$ protein and also indicates that the hippocampus expresses several subtypes of lactate receptors coupled to different signaling effectors [16, 17, 24].

Once activated, the G-protein-coupled receptors can dissociate into $G_\alpha$ and free $G_{\beta\gamma}$ subunits. The $G_{\beta\gamma}$ complex is a potent stimulator of different ion channels and signaling cascades involved in synaptic plasticity [26]. Therefore, we explored the participation of the $G_{\beta\gamma}$ complex in lactate-mediated synaptic potentiation. First, we inhibited the interaction of $G_{\beta\gamma}$ with PLC or IP3-K by including the $G_{\beta\gamma}$ subunit inhibitor gallein (10 µM) in the patch pipette. Intracellular dialysis of gallein blocked synaptic potentiation (RC EPSC at 35 min lactate washout = 96.3 ± 4.12% of baseline response, n = 6; Fig 6A). Interestingly, gallein did not interfere with the increase in the PPR during lactate perfusion (RC PPR control = 1.38 ± 0.04, during lactate perfusion = 1.89 ± 0.13, p<0.05; during washout = 1.32 ± 0.05). Because $G_{\beta\gamma}$ complex activates the IP3-K signaling cascade [27, 28], we next explored the participation of IP3-K. In neurons somatically dialyzed with the IP3-K inhibitor wortmannin (10 µM), perfusion of lactate (2 mM) failed to trigger synaptic potentiation (RC EPSC at 35 min lactate

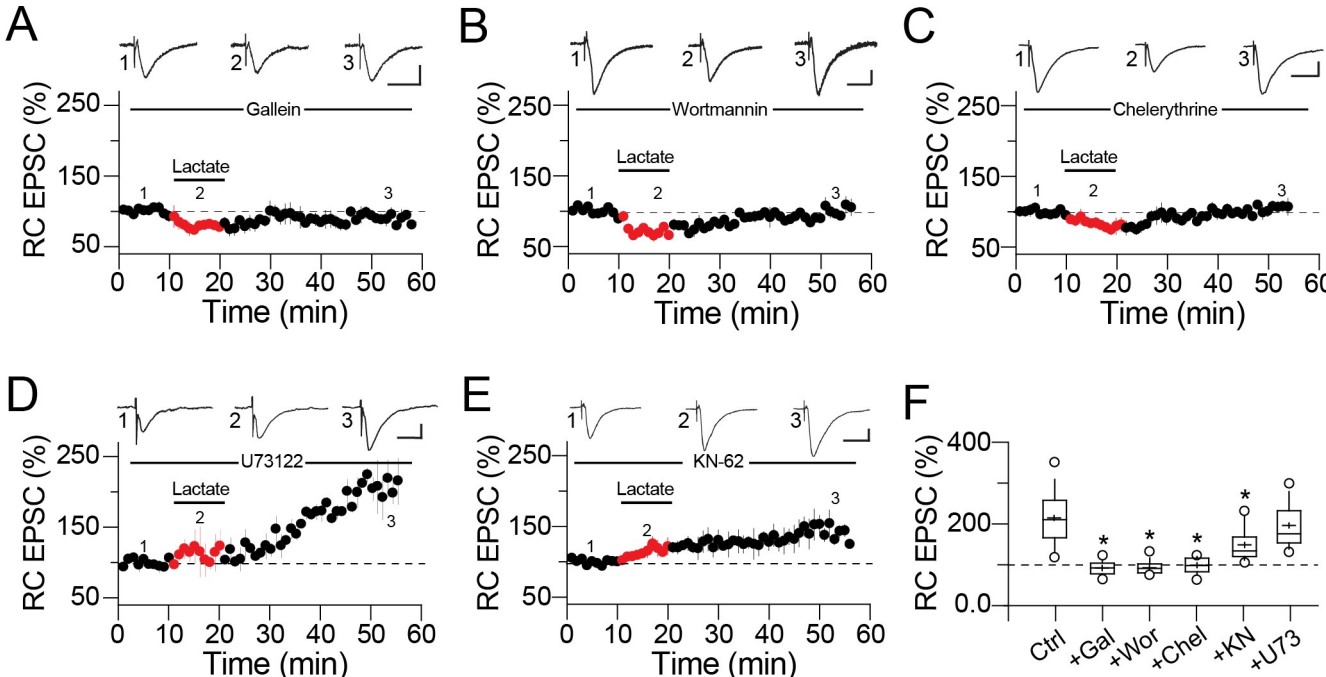

**Fig 6. Intracellular signaling cascades required for synaptic potentiation.** Averaged time-course graphs and representative traces showing the effects of different intracellular signaling cascade blockers. The drugs were included in the patch pipette and drug dialysis lasted 10–15 min before experimental manipulation. **A)** Inclusion of gallein (10 µM; $G_{\beta\gamma}$ subunits inhibitor), **B)** wortmannin (10 µM; IP3-K inhibitor), **C)** chelerythrine (10 µM; PKC inhibitor), **D)** U73122 (10 µM; PLC inhibitor), and **E)** KN-62 (10 µM; CaMKII inhibitor). Whereas synaptic potentiation was insensitive to blockade of PLC, the response showed a partial sensitivity to the postsynaptic blockade of CaMKII. **F)** Boxplots summarizing the magnitude of synaptic potentiation induced with lactate (2 mM) in PC loaded with the indicated drugs.

washout = 92.67 ± 2.6% of baseline response, n = 5; Fig 6B). Cells loaded with wortmannin and exposed to lactate exhibited increased RC PPR (RC PPR control = 1.35 ± 0.06; during lactate perfusion = 1.86 ± 0.05, p < 0.01; during washout = 1.39 ± 0.05; S2C Fig). Since the IP3-K cascade can activate PKC in a PLC-dependent or -independent manner, we also explored the specific contribution of these pathways. Intracellular dialysis of the PLC inhibitor, U73122 (10 μM) did not interfere with either synaptic potentiation or the transient increase in the RC PPR (RC EPSC in cells loaded with U73122 at 35 min lactate washout = 196 ± 9.1%, p > 0.05, n = 4; Fig 6E. RC PPR control = 1.38 ± 0.05; during lactate perfusion = 1.98 ± 0.13, p < 0.01; during washout = 1.4 ± 0.05; S2D Fig). Contrary to this observation, intracellular dialysis of chelerythrine (10 μM) abolished the synaptic potentiation induced with lactate (RC EPSC at 35 min of lactate washout = 107.6 ± 3.5%, n.s., n = 9; Fig 6C). These data indicate that activation of PKC, mediated by the $G_{\beta\gamma}$ complex, is independent of PLC activity and further production of diacylglycerol.

Because CaMKII activity is a necessary step for induction of NMDAR-dependent LTP at the RC synapse [29], we evaluated the participation of CaMKII activity in the synaptic potentiation mediated by lactate. For this, we loaded CA3 PC with the CaMKII blocker, KN-62 (10 μM). Interestingly, dialysis of KN-62 partially reduced the synaptic potentiation magnitude (RC EPSC in KN-62 loaded cells = 148.6 ± 6.03% of baseline response, p < 0.05, n = 4; Fig 6E). The partial reduction of synaptic potentiation when CaMKII is blocked suggests a limited contribution of this signaling cascade during induction of this form of synaptic plasticity. The boxplots (Fig 6F) summarize the effects of the different kinase inhibitors on the synaptic potentiation measured at 35 min of lactate washout. Also, the contribution of the signaling cascades required for induction of synaptic potentiation mediated by lactate is summarized in the proposed model (Fig 8).

## Lactate increases the EPSP-to-spike coupling of CA3 pyramidal cells

If the strengthening of a synapse is accompanied by increased probability of spike generation predicted by a repetitive EPSP discharge (EPSP-to-Spike coupling), then the transient increase in extracellular lactate should augment, in a synapse-specific manner, the spike probability of CA3 PC. According to our results, the increase in the spike probability should be restricted to the RC synapse. To test this prediction, we evoked trains of RC EPSP or MF EPSP (20 or 40 Hz; 20 trains at 10-s intervals) in CA3 PC maintained in current-clamp mode. To minimize that repetitive fiber´s stimulation increases the spike probability, we first constructed input-output relationships to determine a stimulation range that elicited subthreshold responses during train stimulation. The curves were constructed by plotting the RC EPSP or MF EPSP amplitude as a function of stimulus intensity, and the stimulation was set to 30–40% of the subthreshold amplitude response. For the RC synapse, the stimulation current delivered to the SR was 305 ± 15 μA / 100 μs duration; for the MF synapse, stimulation was 400 ± 50 μA μA / 100 μs duration (S1E and S1F Fig). These stimulation ranges were kept constant throughout these experiments. Under control conditions, repetitive RC stimulation at 20 Hz elicited a train of EPSPs without spikes (RC spike probability = 0.83 ± 0.7%; Fig 7A, left panel). As predicted, the spike probability was dramatically increased after lactate perfusion and washout (RC spike probability at 35 min lactate washout = 45.5 ± 15%; Fig 7A, right panel and Fig 7D). The spike probability increase was also observed when stimulation frequency was set to 40 Hz (RC spike probability at 40 Hz = 44.6 ± 14%; n = 6, p < 0.05, Mann-Whitney U test; Fig 7A, lower panels and Fig 7D). The raster plot (Fig 7C) summarizes the increased spike probability of the RC synapse after the synaptic potentiation mediated by lactate.

Unlike the RC-mediated responses, MF-mediated responses are characterized by a strong frequency-dependent facilitation [30]. This feature was noticeable under control conditions, since stimulation at 20 or 40 Hz caused a sustained increase of the MF EPSP amplitude that

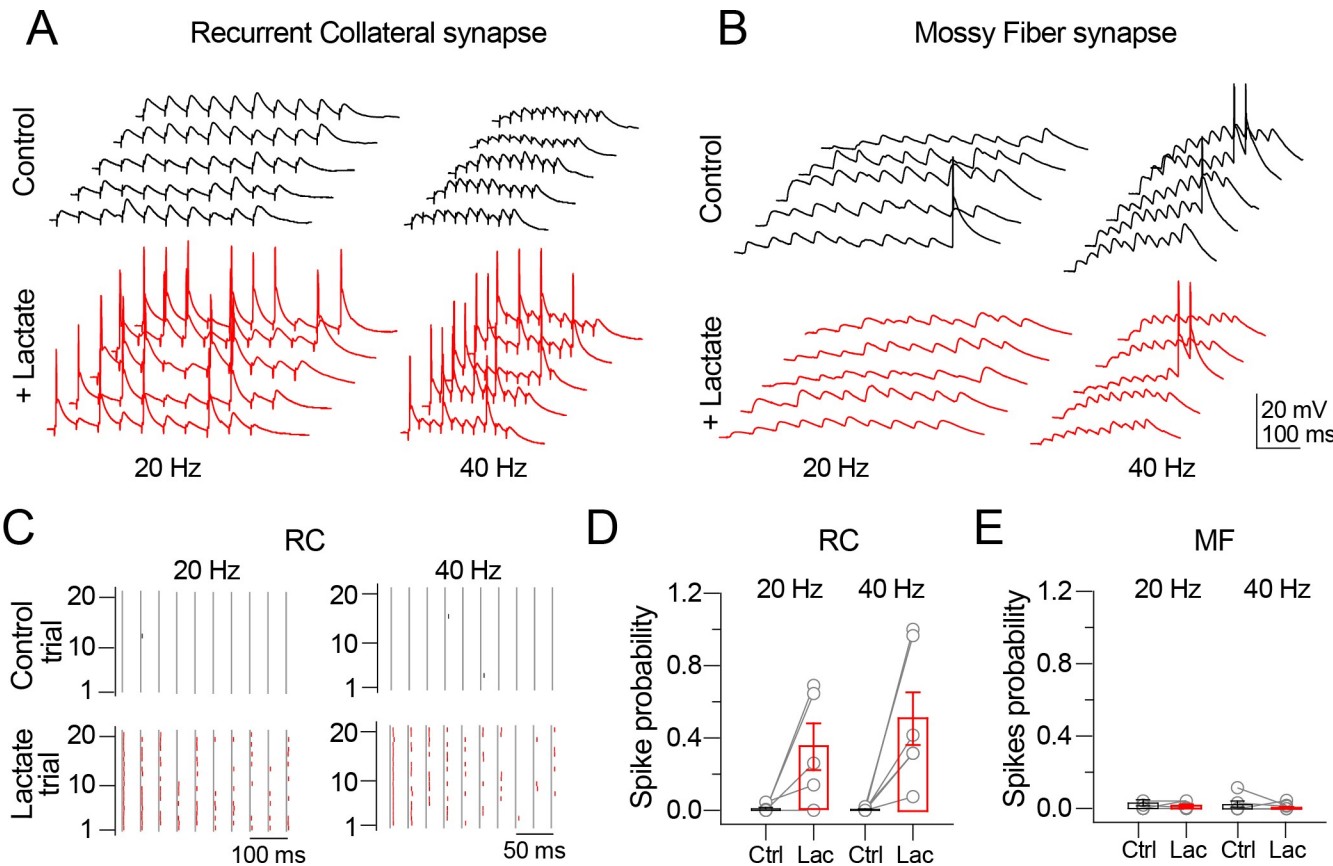

**Fig 7. Lactate triggers synapse-specific increase in spike probability on CA3 PC.** Stimulation trains applied to RC or MF synapses were performed at 20 and 40 Hz for each experimental condition. **A)** Representative trains of RC EPSP before and 30 min after lactate perfusion, evoked at 20 Hz (left panel) and 40 Hz (right panel). Lactate perfusion increased the spike incidence in the RC EPSP train (red traces). **B)** Representative trains of MF EPSP before and 30 min after lactate perfusion, evoked at 20 and 40 Hz. Lactate did not increase the spike incidence on the synaptic response. **C)** Raster plot constructed from the cell depicted in panel A1. Lactate increases the number of spikes (lower panels, red traces). **D–E)** Bar graphs showing the spike probability in control conditions (solid bar) and in the presence of lactate (red bar) for RC and MF synapses. The gray symbols represent individual experiments and the bars the averaged response.

occasionally yielded spikes (see control traces acquired at 20 and 40 Hz, Fig 7B). As predicted, lactate perfusion did not increase the probability of spike generation (control MF spike probability at 20 Hz = 0.79 ± 0.8%; 35 min after lactate washout = 0.28 ± 0.57%. Control spike probability at 40 Hz = 1.7 ± 0.77%; 35 min after lactate washout = 0.92 ± 0.8%; n = 6; n.s., Mann-Whitney U test; Fig 7B and 7E). Likewise, it was noticeable that lactate perfusion and washout caused a mild decrease in the facilitated responses elicited by repetitive MF stimulation, as illustrated in the voltage responses of Fig 7B. Together, these experiments demonstrate that a transient rise in extracellular lactate increases the spike probability of CA3 PC, a phenomenon that occurs in a synapse-specific manner.

A summary of results, including the contribution of the signaling cascades required for induction of synaptic potentiation mediated by lactate in the RC synapse of CA3 PC is depicted in the proposed model (Fig 8).

## Discussion

A transient increase in extracellular lactate triggers synaptic potentiation of the glutamatergic transmission in hippocampal CA3 PC. Since potentiation is restricted to the recurrent

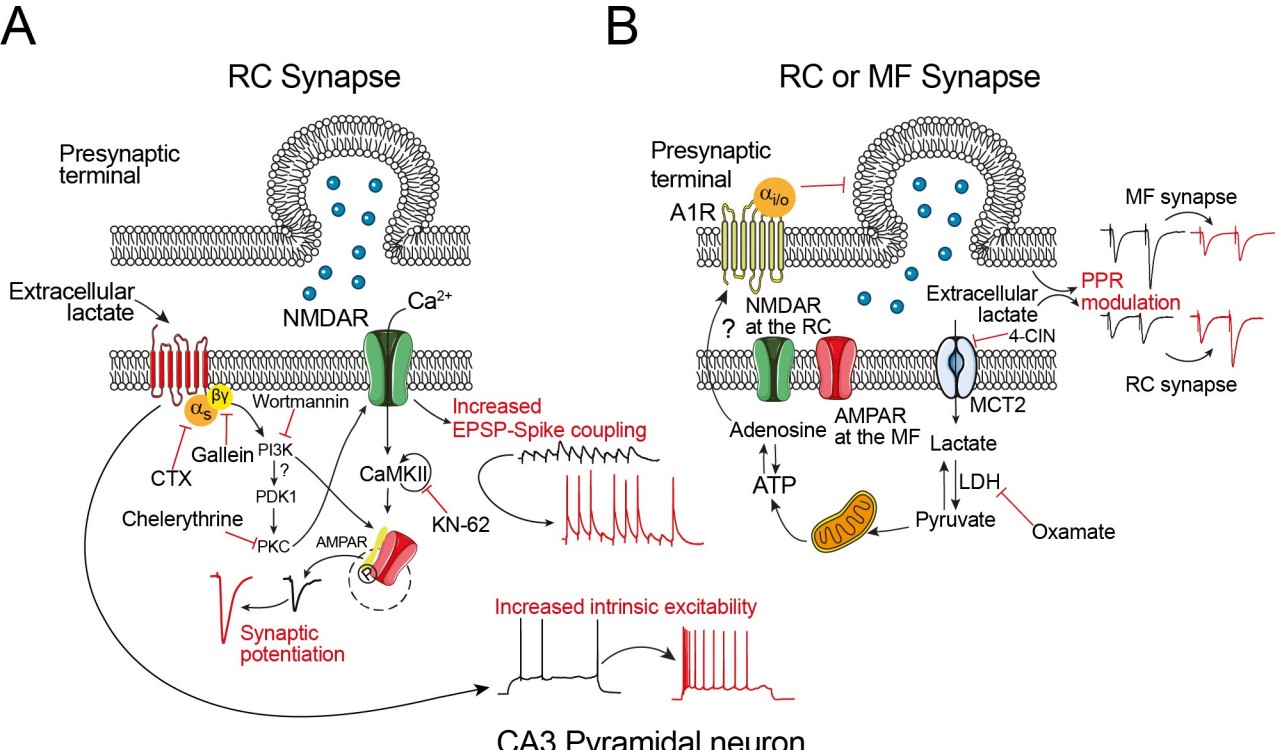

**Fig 8. Proposed mechanism of lactate at the excitatory synapses of CA3 pyramidal cells. A)** Extracellular lactate activates a transmembrane $G_s$ protein-coupled receptor and promotes dissociation of $\alpha$ and $\beta\gamma$ subunits. The $\beta\gamma$ subunits activate the inositol-1,4,5-trisphosphate 3-kinase (IP3-K) pathway and promotes activation of protein kinase C (PKC), probably through PDK1, a rich pleckstrin homology domain-residue protein that responds to PIP3. Activation of IP3-K may facilitate the incorporation of AMPA receptors on the synaptic terminals. It is likely that the active form of PKC also phosphorylates the NMDAR and promotes $Ca^{2+}$ entry and CaMKII to enhance the synaptic strength of the RC synapses. At the RC synapse, lactate increases the EPSP-to-spike coupling. The identity of the ionic conductances underlying the EPSP-to-spike coupling in response to lactate remains to be determined. **B)** Presynaptic modulation of lactate at the RC and MF synapses. Upon its accumulation, lactate is transported to the intracellular milieu via the membranal monocarboxylate transporter 2 (MCT2) and metabolized to pyruvate by the enzyme lactate dehydrogenase (LDH). Then, acetylCoA derived from pyruvate enters the Krebs cycle to produce ATP and indirectly, adenosine. Because RC and especially MF terminals [31] express the inhibitory adenosine A1 receptor, activation of the A1 receptor may be responsible for the depression of the synaptic transmission observed during lactate perfusion. Additionally, astrocytes as prime sources of extracellular ATP and lactate signaling [13] may be involved in the synaptic plasticity of CA3 PC; however further experiments are required to prove this hypothesis. Bottom center: At the postsynaptic level, perfusion of lactate increases the action potential discharge of CA3 PC, by a mechanism sensitive to cholera toxin. The underlying conductances modulated by extracellular lactate operating the increased intrinsic excitability remains to be determined.

collateral synapses and absent in the mossy fibers, this novel form of synaptic plasticity is synapse specific. Beyond its well-known metabolic function, here we have demonstrated that lactate, acting via a postsynaptic receptor, enhances the synaptic strength in a preparation that maintains its neuronal connectivity.

## Lactate requirements for synaptic potentiation on CA3 PC

Our experimental manipulations showed that lactate triggers pre- and postsynaptic activity. On the one hand, the changes in PPR and $CV^{-2}$ imply a presynaptic control of the glutamatergic transmission at both excitatory synapses. On the other hand, the absence of potentiation when the postsynaptic, voltage-operated channels were silenced combined with the pharmacologic manipulations that blocked activation of the intracellular signaling cascades, indicate that lactate also requires postsynaptic activity for synaptic potentiation.

Are the effects of lactate on the RC synapses related to long-term synaptic potentiation? Our experiments demonstrated additional enhancement of the RC EPSC following induction

of electrically-evoked potentiation. Because lactate triggered an attenuated, but significant RC EPSC amplitude increase, we conclude that lactate partially shares cellular mechanisms involved in the electrically-evoked RC potentiation. Several reports have demonstrated that lactate increases the NMDAR currents [32], favor postsynaptic calcium accumulation [18], and promote CaMKII, PKC, and ERK ½ activity [19]. These transductional mechanisms participate in the induction of hippocampal LTP. Lastly, although our occlusion experiments indicate that lactate does not take part in MF LTP, the partial depression mediated by lactate is indicative of a transient control of glutamate release from the mossy boutons, a subject that requires additional investigation.

A growing body of evidence indicates that extracellular lactate modulates multiple electrophysiologic properties [15, 17, 33], neurotransmitter release [18], and controls the intracellular calcium concentration [34, 35]. Consistent with this evidence, our results support the hypotheses that lactate operates as a 'gliotransmitter' [18, 36–38] to modulate the synaptic strength converging on CA3 PC. What is more, building on earlier evidence indicating that memory formation requires lactate metabolism [12, 13], our results uncovered a novel mechanism the involves activation of postsynaptic receptors and activation of intracellular cascades by which lactate controls the strength of the synaptic transmission underlying the mnemonic processing of the hippocampal formation.

Our experiments also revealed that lactate has a biphasic action on synaptic transmission. First, depression of the glutamatergic transmission, followed by potentiation (at the RC synapse) or a transient depression (at the MF synapse). Because gamma oscillations mediated by GABAergic interneurons in the hippocampal area CA3 are modulated by lactate and the ketone body β-hydroxybutyrate [39, 40], the partial depression observed in the glutamatergic transmission may result from modulation of the GABAergic transmission. However, this possibility is unlikely since our manipulations were performed in the presence of PTX. On the other hand, when hippocampal slices were preincubated with oxamate to prevent the conversion of lactate to pyruvate and vice versa, both the partial depression and changes in the PPR were indeed prevented (see Fig 4A). One possibility to explain these results is that lactate, acting via a metabolic-related mechanism, is able to control and decrease glutamate release from the presynaptic terminal. In support of this idea, we also found that pyruvate transiently modifies the PPR. The metabolic-related effect of lactate is an interesting subject of research that demands further investigation.

Mechanistically, we found that potentiation requires coincident postsynaptic depolarization and voltage-operated channel availability during lactate elevation. Indeed, the lack of potentiation observed when cells were held at -100 mV or loaded with BAPTA, or when the NMDAR were blocked, indicates that calcium elevation is a necessary step for synaptic potentiation. These data also suggest that the primary source of calcium entry is via activation of NMDAR. This plausible mechanism is reminiscent of the electrically evoked, NMDAR-dependent LTP at the RC synapses of CA3 PC [41, 42]; interestingly, this process also requires activity of CaMKII [43] and PKC [44–46].

## CA3 pyramidal cells express an atypical lactate receptor

Another prominent finding is that lactate activates postsynaptic receptors sensitive to 3,5-DHBA, the specific HCA1 receptor agonist [47]. However, 3,5-DHBA did not modify the RC PPR or $CV^{-2}$. The lack of changes in PPR and $CV^{-2}$, and the sensitivity to cholera toxin rather than pertussis toxin, indicate that CA3 PC express a lactate-sensitive receptor coupled to a different effector system. In primary cortical neurons, it was demonstrated that the HCA1 receptor signals via a $G_\alpha/G_{\beta\gamma}$ system. However, activation of cortical HCA1 with lactate or

3,5-DHBA decreases the frequency of miniature EPSC, intracellular calcium concentration, and action potential discharge [48]; see also [15], discarding the possibility that the lactate-sensitive receptor of CA3 PC is the same as that expressed by primary cortical neurons. Another study reported a lactate-sensitive receptor that increases both the intracellular calcium and firing discharge of locus coeruleus neurons. These authors proposed the existence of a variant of the HCA1 receptor that couples to a different signaling cascade, or the existence of another receptor that recognizes lactate [18]. However, the identification of the receptor expressed by hippocampal CA3 PC requires further investigation. Our results also open up the possibility that different lactate-sensitive receptors coupled to multiple signaling pathways can be expressed on the different neuronal populations of the hippocampus.

Another intriguing observation is the modulatory action of lactate on MF-mediated synaptic transmission and MF LTP. If HCA1 is expressed at the presynaptic terminals (as indicated by the changes in PPR, $CV^{-2}$ and transient depression during expression of MF LTP), why does lactate not trigger presynaptic MF potentiation? We found two possibilities that may account for this disparity. One possibility involves the fact that HCA1 is negatively coupled to the adenylyl cyclase-cAMP-PKA system, which is a necessary step for the induction of MF LTP [49]. It is possible that the negative coupling of the lactate receptor to this signaling cascade restrains the induction of potentiation at the MF synapse while facilitating potentiation at the RC terminal. The other possibility involves HCA1 crosstalk with the adenosine A1 receptor [48]. This interaction results in decreased excitability. If this is the case, increased activation of the A1 receptor following activation of the lactate receptor would reduce the excitability at the giant mossy bouton terminals and prevent presynaptic MF potentiation [31].

## Lactate controls the output of CA3 pyramidal cells

Notably, the output of CA3 PC increased in response to lactate. This phenomenon was observed both at the synaptic but also at the somatic level. Even though the response is the same, i.e., increased AP discharge, the triggering mechanism is quite different. On the one hand, our data show that lactate increases the EPSP-to-spike coupling, a phenomenon restricted to the RC synapse. On the other, we found increased intrinsic excitability in response to somatic injection of current. The former involves pre and postsynaptic modulation of neurotransmitter receptors and perhaps ion channels, whereas the latter involves somatic modulation of the ionic conductances that shape the PC output. A previous report [50] demonstrated that somatic firing of CA3 neurons downregulates the D-type potassium channels (Kv1.2) located at the distal apical dendrites, a mechanism dependent on $Ca^{2+}$ and protein tyrosine kinases (PTK). Downregulation of Kv channels promotes increased somatic excitability. Since lactate promotes $Ca^{2+}$ accumulation to induce synaptic potentiation, lactate may activate a PTK signaling mechanism that, in turn, would promote changes in the expression of ionic channels located at the distal synapses of CA3 PC, i.e., recurrent collateral synapses. If this is the case, it remains to be demonstrated if lactate exerts dendritic modulation of potassium channels, as a plausible mechanism underlying the EPSP-to-spike coupling. On the other hand, a growing number of works demonstrated that lactate modulates multiple ion channels and modifies the firing discharge and intrinsic excitability [15, 17, 48, 51]. Thus, it is reasonable to assume that lactate may exert specific actions on one synaptic input (RC) and simultaneously exert modulatory actions on the ionic conductances that shape the action potential discharge.

## Physiologic implications

In the hippocampus, the extracellular lactate concentration fluctuates in the low millimolar range, between 1 to 1.4 mM [3, 52–55]; and physiologic activity, such as activation of the

entorhinal cortex or astrocytic stimulation, induces lactate release that develops within seconds of neural activation [4, 37]. Given that the predominant excitatory network of area CA3 arises from the recurrent axons of CA3 PC [23, 42, 56, 57], and computational models point to the RC circuitry as the auto-associative network that retrieves complete information on the basis of incomplete retrieval cues stored in hippocampal area CA3 (or pattern completion, see [58, 59]), we hypothesize that the transient rises in extracellular lactate favors the mnemonic process of pattern completion by strengthening the RC synapses and facilitating the output of CA3 PC without compromising the synaptic transmission mediated by the dentate mossy fibers.

## Supporting information

**S1 Fig. A)** Average time-course graph showing the effects of lactate 2 mM on the RC EPSC acquired in an artificial cerebrospinal fluid (aCSF) containing glucose 5 mM. Under this condition, the passive and active electrophysiologic properties of CA3 PC were similar to the recorded in the standard aCSF containing glucose 10 mM. The synaptic potentiation triggered by lactate (2 mM, 10 min) was similar to the potentiation reported in Fig 1D. Inset, representative RC EPSC obtained in control conditions (1) and 30 min after lactate washout (2). **B)** Average time-course graph showing the effects of removing glucose on the normalized RC EPSC amplitude acquired in an aCSF containing glucose 5 mM. As previously reported (Schurr et al., 1988; Tecuatl et al., 2018), removing glucose irreversibly suppressed the RC EPSC. Upper inset, representative RC EPSC obtained in control conditions (1) and 30 min after removing glucose (2). The suppression of the evoked responses on CA3 PC was accompanied by propidium iodide accumulation, indicating reduced cell viability. **C-D)** Higher magnifications of confocal microphotographs showing the PI uptake in slices maintained in aCSF containing glucose 5 mM or low glucose concentration. PI accumulation (red staining) was observed in slices containing as low as 2 mM of glucose. The details for propidium iodide uptake and fluorescence microscopy on acute hippocampal slice has been previously published by our group (Tecuatl et al., 2018). **E-F)** Scatter plots of current intensity vs. EPSP amplitude acquired in current-clamp mode at RMP of CA3 PC. The input-output curves were used to determine a stimulus that caused 30–40% of the subthreshold response to avoid saturation (spike generation) following repetitive stimulation (20 stimuli at 20 or 40 Hz at 10-s intervals; See Fig 8). RC synapses required less current injection to generate spikes, whereas MF stimulation triggered larger synaptic responses before spike onset. The gray areas on the curve represent the injected current ± SEM used to stimulate each CA3 synapse.
(TIF)

**S2 Fig. Modulation of the paired-pulse ratio at the recurrent collateral synapses of CA3 PC during different pharmacologic manipulations. A)** Extracellular perfusion of the non-competitive inhibitor of the lactate dehydrogenase, oxamate (10 mM, 20 min), prevented the increase in the PPR during lactate perfusion. **B)** Perfusion of the monocarboxylate transporter-2 blocker, 4-CIN (0.5 mM) did not alter the PPR at the RC synapse of CA3 PC but prevented its increase during lactate perfusion. **C)** Somatic loading with the IP3-K signaling cascade blocker, wortmannin (10 μM), does not interfere with the modulation of PPR by lactate. Likewise, **D)** Somatic loading with the inhibitor of the Protein lipase C (PLC), U73122 (10 μM) does not interfere with the increase in PPR exerted by perfusion of lactate. For all the bars (Mean ± SEM), each symbol represents an independent experiment.
(TIF)

## Acknowledgments

This work was acquired with data from Gabriel Herrera-Lopez' Ph.D. dissertation thesis. The authors thank professors Hubert Fiumelli and Germán Barrionuevo for insightful comments during the preparation of the manuscript.

## Author Contributions

**Conceptualization:** Gabriel Herrera-López, Emilio J. Galván.

**Data curation:** Gabriel Herrera-López, Emilio J. Galván.

**Formal analysis:** Gabriel Herrera-López.

**Funding acquisition:** Emilio J. Galván.

**Investigation:** Gabriel Herrera-López, Ernesto Griego, Emilio J. Galván.

**Methodology:** Gabriel Herrera-López, Ernesto Griego, Emilio J. Galván.

**Project administration:** Emilio J. Galván.

**Resources:** Emilio J. Galván.

**Software:** Emilio J. Galván.

**Supervision:** Emilio J. Galván.

**Validation:** Emilio J. Galván.

**Writing – original draft:** Gabriel Herrera-López.

**Writing – review & editing:** Emilio J. Galván.

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
