## [Decision Letter · Decision Letter 0]

26 Aug 2020

PONE-D-20-21322

Lactate induces synapse-specific potentiation on CA3 pyramidal cells of rat hippocampus

PLOS ONE

Dear Dr. Galvan,

Thank you for submitting your manuscript to PLOS ONE. After careful consideration, we feel that it has merit but does not fully meet PLOS ONE’s publication criteria as it currently stands. Therefore, we invite you to submit a revised version of the manuscript that addresses the points raised during the review process.

Your manuscript has been reviewed by two independent experts whose comments can be found below. Both experts found your study to be novel, well conducted and merits to be published but raised some concerns. I do agree with their comments notably regarding the necessity to discuss the mechanisms beyond the compartimentalisation of the effects of lactate on synaptic transmission and on cell excitability. I also agree that you may consider to use non parametric tests for your analysis rather than Student t test when size of samples is too small.

We look forward to receiving your revised manuscript.

Kind regards,

Jean-Pierre Mothet, Ph.D

Academic Editor

PLOS ONE

Journal Requirements:

2.  Thank you for including your ethics statement: 'The Ethics Committee for Animal Research of our institution approved all experimental procedures according to Mexican legislation (NOM-062-ZOO-1999; authorization number 0090-14), which were performed in adherence to the “Guide for the care and use of Laboratory Animals (NIH 8th edition, 2011).'

a. Please amend your current ethics statement to include the full name of the ethics committee that approved your specific study.

For additional information about PLOS ONE submissions requirements for ethics oversight of animal work, please refer to http://journals.plos.org/plosone/s/submission-guidelines#loc-animal-research

Reviewers' comments:

Reviewer's Responses to Questions

**Comments to the Author**

1. Is the manuscript technically sound, and do the data support the conclusions?

Reviewer #1: Yes

Reviewer #2: Partly

2. Has the statistical analysis been performed appropriately and rigorously? 

Reviewer #1: No

Reviewer #2: Yes

3. Have the authors made all data underlying the findings in their manuscript fully available?

Reviewer #1: Yes

Reviewer #2: Yes

4. Is the manuscript presented in an intelligible fashion and written in standard English?

Reviewer #1: Yes

Reviewer #2: Yes

5. Review Comments to the Author

Reviewer #1: This is a very nice study that examines the effects of lactate on synaptic efficacy and intrinsic excitability. Using electrophysiological techniques on acute slices of rat hippocampus, the authors convincingly show that application of lactate persistently enhances synaptic transmission at recurrent synapses but not at mossy fiber synapses in CA3 pyramidal neurons. Lactate-dependent LTP at recurrent synapses requires NMDA receptor activation, postsynaptic calcium and is sensitive to cholera toxin. The lactate induces in parallel an increase in cell excitability that is surprisingly visible on recurrent inputs but not on mossy-fiber inputs.

The question raised here is of importance, the data are clear and the paper is well written. I have however a few questions and remarks.

Excitability change illustrated in Figure 5H-J: Is it an acute effect of lactate or a long-term effect measured in parallel of synaptic transmission? Please provide details in the Methods & Results sections.

How the authors can reconcile the fact that lactate induces an increase in intrinsic excitability measured by current injection in the soma and the differential effect on recurrent and mossy-fiber inputs? A change in excitability that is global should affect equally all inputs and especially inputs located near the cell body (i.e. MF inputs). A selective change in EPSP-spike coupling can be seen only in specific conditions (see for instance Campanac et al. J Neurosci 2008).

Although, it is mentioned in the abstract, the effect of lactate on intrinsic excitability is not discussed. Please add a brief paragraph on this after clarification of the synapse specific change in excitability with at least a reference to a recent review.

Statistics: Student t-test should not be used for small samples. Please use non parametric tests (Mann-Whitney and Wilcoxon tests).

Reviewer #2: In their paper “Lactate induces synapse-specific potentiation on CA3 pyramidal cells of rat Hippocampus”, Herrera-Lopez et al. report input source-dependent potentiation of synaptic transmission onto hippocampal C3 pyramidal cells by lactate, and suggest that this effect is mediated by a Gs-protein-coupled lactate receptor. Synaptic potentiation applies only to recurrent C3 synapses but not to mossy fiber synapses. In addition, lactate causes transient synaptic depression presumably via a metabolism-related effect. To note, a physiologically relevant range of lactate concentrations was tested. This is very exciting new data and substantiated by electrophysiological and pharmacological approaches in mature rodent tissue.

The text is well written, with very few typographical and syntax errors (listed under specific comments below). Description of methods is mostly transparent, apart from some aspects of the electrical stimulation protocol which I would ask to clarify in more detail (see specific comments). With a few exceptions (see specific comments), the figures are well designed and clearly described in results section as well as legends.

My main concern is that the mechanistical interpretation of the differences between lactate effects on RC and MF transmission requires further development. The Discussion and Figure 8 appear to ascribe the lactate-induced synaptic potentiation in RC, but not MF, synapses to postsynaptic changes. However, since the postsynaptic cell is identical (a CA3 PC), the logic isn’t obvious, unless the authors would want to explain this difference by compartmentalisation and local signalling of the G-protein-coupled lactate receptor? Their reasoning should be explained in more detail in the Discussion to include lactate actions on MF as well as RC synapses, and this should be reflected by Figure 8.

Specific comments:

Line 92: … and then kept for 90 min…

Line 123 etc – Stimulation protocol unclear: What does “70 ms at 0.06 Hz” mean and how do these parameters relate to the “tandem analysis” introduced in line 161 as “…acquired with an ISI of 1 s and stimulation frequency of 0.16 Hz”? Also see legend to Fig 1.

The time course plots in the Figures (e.g. 1B) suggest a stimulation frequency of ~0.02 Hz for each pathway. If “tandem” means alternating stimulation between MF and RC, I would expect them to be applied in equal ~25 sec intervals, in order for activation of one pathway not to bias the response of the other. Please clarify in the Methods section what exactly was done.

Also, it is not explained how CV^-2 (first appearance line 234) was determined and that should be done in order to ensure accessibility to a wider readership.

Lastly, one assumes that what is reported are normalised EPSC amplitudes but that isn’t explicitly stated in the manuscript.

Line 137 – Drugs: Lactate not mentioned; please also state pH of stock solution.

General: Inconsistent use of abbreviation for pyramidal cells, suggest to introduce at first instance in main text and then use throughout.

Lines 209,211: Because… Because (suggest to replace one with “Since”)

Lines 213-14: “…capable of potentiating the synaptic…”

Line 216: “…EPSC to a similar extent…”

Fig 2D: LTP label – why “LTP”? this should probably be representing “synaptic potentiation”.

Line 314: “… of CA3 PC have a predominant NMDAR component involved in RC long-term…”

Line 361-362: “…of lactate. Although pyruvate (2 mM) did not induce RC EPSC potentiation, it did

increase, in a transient manner, the RC…”

Lines 400 etc: Legend to Figure 4B is missing (and much needed as the results text doesn’t make it very clear what was done here) – the text to B) seems to refer to panel C, etc.

Fig 5B: LTP label – should say 3,5-DHBA? In this case, the figure represents “synaptic potentiation” but not LTP (as in Figs 3, 7)?

Line 646: “…Lastly, although our occlusion experiments…”

Lines 724 – 729: Sentence incomplete

6. PLOS authors have the option to publish the peer review history of their article (what does this mean?). If published, this will include your full peer review and any attached files.

Reviewer #1: No

Reviewer #2: No

---

## [Author Response · Author response to Decision Letter 0]

5 Oct 2020

Responses to the reviewers

Journal Requirements

1. PLOS ONE's style requirements: 

• We have double check the style requirement. 

2. Ethics statement:

• We have amended this statement and the full name of the ethics committee, and approbation number for our study is included. 

3. Data are available upon request:

• Since we have no legal or ethical restrictions, we have uploaded the data to a public repository (link: https://gin.g-node.org/gabriel.helo/Lactate-potentiation).

4. Phrase “data not shown”:

• We have included the actual values for each result referred as “not shown”. Indeed, the result section is upgraded with new numeric values, also shown in a new supplementary Fig 2 that summarizes these findings. 

5. Captions for your Supporting Information at the end of the manuscript:

• The supporting Figs captions are included at the end of the manuscript.

Reviewer #1: 

Excitability change illustrated in Figure 5H-J: Is it an acute effect of lactate or a long-term effect measured in parallel of synaptic transmission? Please provide details in the Methods & Results sections.

• Experiments assessing changes in the intrinsic excitability were acquired at least at three different moments (in agreement with our previous publication Herrera-López and Galván, 2018). Measurements were performed during lactate perfusion, between 10-15 min after lactate washout and ≈30-35 min after lactate washout. We found that the firing discharge measured at 15 and 35 min were comparable; however, we did not use the latter values, since the intracellular dialysis of the whole-cell configuration could affect our results. Thus, we restricted our measurements to ≈13-15 min after lactate washout. We have incorporated an explanatory paragraph in the material and methods section to clarify this issue.

How the authors can reconcile the fact that lactate induces an increase in intrinsic excitability measured by current injection in the soma and the differential effect on recurrent and mossy-fiber inputs? A change in excitability that is global should affect equally all inputs and especially inputs located near the cell body (i.e. MF inputs). A selective change in EPSP-spike coupling can be seen only in specific conditions (see for instance Campanac et al. J Neurosci 2008). Although, it is mentioned in the abstract, the effect of lactate on intrinsic excitability is not discussed. Please add a brief paragraph on this after clarification of the synapse specific change in excitability with at least a reference to a recent review.

• Thanks for this comment. We have added a new paragraph in the discussion addressing this specific issue. Briefly, we discuss the differences between the intrinsic excitability vs the increased EPSP-to-spike coupling observed at the RC synapses. We also included several references that support our hypothesis (highlighted in the new version of the manuscript, discussion).

Statistics: Student t-test should not be used for small samples. Please use non parametric tests (Mann-Whitney and Wilcoxon tests).

• The appropriate statistical analyses are incorporated in the new version of the manuscript (highlighted text). 

Reviewer #2: In their paper “Lactate induces synapse-specific potentiation on CA3 pyramidal cells of rat Hippocampus”, Herrera-Lopez et al. report input source-dependent potentiation of synaptic transmission onto hippocampal C3 pyramidal cells by lactate, and suggest that this effect is mediated by a Gs-protein-coupled lactate receptor. Synaptic potentiation applies only to recurrent C3 synapses but not to mossy fiber synapses. In addition, lactate causes transient synaptic depression presumably via a metabolism-related effect. To note, a physiologically relevant range of lactate concentrations was tested. This is very exciting new data and substantiated by electrophysiological and pharmacological approaches in mature rodent tissue.

The text is well written, with very few typographical and syntax errors (listed under specific comments below). Description of methods is mostly transparent, apart from some aspects of the electrical stimulation protocol which I would ask to clarify in more detail (see specific comments). With a few exceptions (see specific comments), the figures are well designed and clearly described in results section as well as legends.

My main concern is that the mechanistical interpretation of the differences between lactate effects on RC and MF transmission requires further development. The Discussion and Figure 8 appear to ascribe the lactate-induced synaptic potentiation in RC, but not MF, synapses to postsynaptic changes. However, since the postsynaptic cell is identical (a CA3 PC), the logic isn’t obvious, unless the authors would want to explain this difference by compartmentalisation and local signalling of the G-protein-coupled lactate receptor? Their reasoning should be explained in more detail in the Discussion to include lactate actions on MF as well as RC synapses, and this should be reflected by Figure 8.

Specific comments:

Line 92: … and then kept for 90 min…

• This change is highlighted in the new version of the manuscript. 

Line 123 etc – Stimulation protocol unclear: What does “70 ms at 0.06 Hz” mean and how do these parameters relate to the “tandem analysis” introduced in line 161 as “…acquired with an ISI of 1 s and stimulation frequency of 0.16 Hz”? Also see legend to Fig 1.

The time course plots in the Figures (e.g. 1B) suggest a stimulation frequency of ~0.02 Hz for each pathway. If “tandem” means alternating stimulation between MF and RC, I would expect them to be applied in equal ~25 sec intervals, in order for activation of one pathway not to bias the response of the other. Please clarify in the Methods section what exactly was done.

• We have added a paragraph (material and methods) to explain this stimulation protocol. Briefly (see also the included schematic representation below), each PClamp sweep lasted 1 sec. Then, we delivered a pair of stimuli on the MF (paired pulse with an interval [ISI] of 70 ms) at 50 ms after the beginning of the sweep, followed by second pair of stimuli on the RC (S. Radiatum). 800 ms after the MF stimulation. The idea of this experiment was to assess the effects of lactate on both glutamatergic synapses of the target cells. A similar stimulation protocol was used in previous works of our laboratory (see Galvan et al., 2015; Villanueva-Castillo et al., 2017) to simultaneously assess pharmacologic effects of drugs on converging synapses of interneurons or pyramidal cells. The tandem protocol is visually explained in the next scheme:

Tandem protocol for MF and RC stimulation. Two independent pathways (MF and RC) were consecutively stimulated with paired current pulses (100 µM). Pairs of stimuli were delivered with an ISI of 70 ms to measure the paired-pulse ratio. To minimize possible synaptic interactions, MF and RC stimuli had a delay of 800 ms while acquired in the same sweep (1 sec duration). The interval between sweeps was 15 sec (0.016 Hz).

Also, it is not explained how CV^-2 (first appearance line 234) was determined and that should be done in order to ensure accessibility to a wider readership.

• We have included a text line and a reference that explain in detail the relevance of the CV^-2 (Highlighted in the new version of the manuscript

Lastly, one assumes that what is reported are normalised EPSC amplitudes but that isn’t explicitly stated in the manuscript.

• Thanks for this comment. We included a short sentence that explicitly states that we measured normalized EPSC amplitudes (statistical analysis section, highlighted)

Line 137 – Drugs: Lactate not mentioned; please also state pH of stock solution.

• We have included a short line with the specifications of the lactate used in the study, as indicated by the reviewer (drugs section of material and methods, highlighted text). 

General: Inconsistent use of abbreviation for pyramidal cells, suggest to introduce at first instance in main text and then use throughout.

• For the new version of the manuscript, we have included the pyramidal cells (CA3 PC) abbreviation at the first sentence of the introduction (highlighted). However, we did not use this abbreviation in the result subheadings and figure legends of the manuscript

Lines 209,211: Because… Because (suggest to replace one with “Since”)

• This line has been amended as suggested by the reviewer. 

Lines 213-14: “…capable of potentiating the synaptic…”

• We thank the reviewer for this observation, we have amended this mistake. 

Line 216: “…EPSC to a similar extent…”

• This line has been amended as suggested by the reviewer.

Fig 2D: LTP label – why “LTP”? this should probably be representing “synaptic potentiation”.

• We agree with the reviewer. The label represents potentiation not LTP and we have updated all the figs where this mistaken label was present.

Line 314: “… of CA3 PC have a predominant NMDAR component involved in RC long-term…”

• This line is amended in the new version of the manuscript. We appreciate this observation.

Line 361-362: “…of lactate. Although pyruvate (2 mM) did not induce RC EPSC potentiation, it did

increase, in a transient manner, the RC…” 

• This change is incorporated in the new version of the manuscript.

Lines 400 etc: Legend to Figure 4B is missing (and much needed as the results text doesn’t make it very clear what was done here) – the text to B) seems to refer to panel C, etc.

• We have amended the labelling of Fig 4. (highlighted) We appreciate this observation. 

Fig 5B: LTP label – should say 3,5-DHBA? In this case, the figure represents “synaptic potentiation” but not LTP (as in Figs 3, 7)? 

• We have amended the labeling for all the figures. As the reviewer indicates, we have synaptic potentiation during lactate washout, not LTP. 

Line 646: “…Lastly, although our occlusion experiments…” 

• This change is incorporated in the new version of the manuscript.

Lines 724 – 729: Sentence incomplete

• This line has been amended in the new version of the manuscript.

---

## [Decision Letter · Decision Letter 1]

2 Nov 2020

Lactate induces synapse-specific potentiation on CA3 pyramidal cells of rat hippocampus

PONE-D-20-21322R1

Dear Dr. Galvan,

We’re pleased to inform you that your manuscript has been judged scientifically suitable for publication and will be formally accepted for publication once it meets all outstanding technical requirements.

Kind regards,

Jean-Pierre Mothet, Ph.D

Academic Editor

PLOS ONE

Additional Editor Comments (optional):

Reviewers' comments:

Reviewer's Responses to Questions

**Comments to the Author**

1. If the authors have adequately addressed your comments raised in a previous round of review and you feel that this manuscript is now acceptable for publication, you may indicate that here to bypass the “Comments to the Author” section, enter your conflict of interest statement in the “Confidential to Editor” section, and submit your "Accept" recommendation.

Reviewer #1: All comments have been addressed

Reviewer #2: (No Response)

2. Is the manuscript technically sound, and do the data support the conclusions?

Reviewer #1: Yes

Reviewer #2: Yes

3. Has the statistical analysis been performed appropriately and rigorously? 

Reviewer #1: Yes

Reviewer #2: Yes

4. Have the authors made all data underlying the findings in their manuscript fully available?

Reviewer #1: Yes

Reviewer #2: Yes

5. Is the manuscript presented in an intelligible fashion and written in standard English?

Reviewer #1: Yes

Reviewer #2: Yes

6. Review Comments to the Author

Reviewer #1: The paper has been adequately revised (statiistics use non-parametric tests, and previous concerns have been dissipated). I have no further comment.

Reviewer #2: I am satisfied with revisions the authors have undertaken, and appreciate the efforts gone into more detailed interpretation of the differential effects of lactate on RC and MF synaptic transmission, including the improvements in Fig 8. The legend would still benefit from some cuts and rephrasing. For example, for a title “Proposed mechanisms of lactate actions at the excitatory synapses of the CA3 pyramidal cells” may be more precise; “Schematic representation.” is unnecessary; the explanation of Gs-protein activation (“…induces a conformational change …”) seems too extensive; pyruvate does not “synthesize ATP”, nor adenosine.

As the authors acknowledge in their revised manuscript, a number of steps in their model will require further clarification through future work. I suggest these would include the currently unexplained mechanism of the proposed post-synaptic release of ATP. In fact, local astrocytes as prime sources of extracellular ATP and lactate signalling may well be involved in the synaptic potentiation processes described in this paper.

7. PLOS authors have the option to publish the peer review history of their article (what does this mean?). If published, this will include your full peer review and any attached files.

Reviewer #1: No

Reviewer #2: No

---

## [Editor Report · Acceptance letter]

4 Nov 2020

PONE-D-20-21322R1 

Lactate induces synapse-specific potentiation on CA3 pyramidal cells of rat hippocampus 

Dear Dr. Galvan:

I'm pleased to inform you that your manuscript has been deemed suitable for publication in PLOS ONE. Congratulations! Your manuscript is now with our production department. 

Kind regards, 

on behalf of

Dr Jean-Pierre Mothet 

Academic Editor

PLOS ONE